# Minimal requirements for a neuron to coregulate many properties and the implications for ion channel correlations and robustness

Jane Yang[1,2], Husain Shakil[1,3†], Stéphanie Ratté[1], Steven A Prescott[1,3]*

[1]Neurosciences and Mental Health, The Hospital for Sick Children, Toronto, Canada; [2]Institute of Biomedical Engineering, University of Toronto, Toronto, Canada; [3]Department of Physiology, University of Toronto, Toronto, Canada

**\*For correspondence:**
steve.prescott@sickkids.ca

**Present address:** †Division of Neurosurgery, Department of Surgery, University of Toronto, Toronto, Canada

**Abstract** Neurons regulate their excitability by adjusting their ion channel levels. Degeneracy – achieving equivalent outcomes (excitability) using different solutions (channel combinations) – facilitates this regulation by enabling a disruptive change in one channel to be offset by compensatory changes in other channels. But neurons must coregulate many properties. Pleiotropy – the impact of one channel on more than one property – complicates regulation because a compensatory ion channel change that restores one property to its target value often disrupts other properties. How then does a neuron simultaneously regulate multiple properties? Here, we demonstrate that of the many channel combinations producing the target value for one property (the single-output solution set), few combinations produce the target value for other properties. Combinations producing the target value for two or more properties (the multioutput solution set) correspond to the intersection between single-output solution sets. Properties can be effectively coregulated only if the number of adjustable channels ($n_{in}$) exceeds the number of regulated properties ($n_{out}$). Ion channel correlations emerge during homeostatic regulation when the dimensionality of solution space ($n_{in} - n_{out}$) is low. Even if each property can be regulated to its target value when considered in isolation, regulation as a whole fails if single-output solution sets do not intersect. Our results also highlight that ion channels must be coadjusted with different ratios to regulate different properties, which suggests that each error signal drives modulatory changes independently, despite those changes ultimately affecting the same ion channels.

## Editor's evaluation

Neurons develop and maintain rich electrophysiological properties that enable nervous systems to function. The question of how different neural properties are regulated by internal ion channel expression mechanisms remains unresolved. In this paper, Yang and colleagues address the question from an abstract perspective by asking how multiple constraints on the physiological properties of neurons, such as firing rate curves and energy efficiency, narrow down the available regulatory possibilities. Their results from a mixture of modelling and dynamic clamp experiments point to the existence of multiple parallel internal feedback loops for controlling ion channel expression in neurons, and derive conditions under which co-regulation mechanisms will fail.

## Introduction

Neurons maintain their average firing rate near a target value, or set point, by adjusting their intrinsic excitability and synaptic weights (*Turrigiano et al., 1994*; *O'Leary et al., 2010*; *Aizenman et al., 2003*; *Desai et al., 1999*; *Hengen et al., 2013*; *van Welie et al., 2006*). Homeostatic regulation of intrinsic excitability is achieved through feedback control of diverse ion channels (*Turrigiano et al., 1994*; *O'Leary et al., 2010*; *Desai et al., 1999*; *Joseph and Turrigiano, 2017*). Computational models have successfully employed negative feedback to adjust ion channel densities (*O'Leary et al., 2014*; *O'Leary and Marder, 2016*; *Liu et al., 1998*; *Olypher and Prinz, 2010*; *LeMasson et al., 1993*) and control theory provides a valuable framework to conceptualize how this occurs (*O'Leary and Wyllie, 2011*). But most of the mechanistic details remain unclear (*Davis, 2006*; *Turrigiano, 2011*) and are not straightforward; for instance, different perturbations can trigger similar changes in excitability via different signaling pathways affecting different ion channels (*Kulik et al., 2019*), or via different combinations of excitability changes and synaptic scaling (*Maffei and Turrigiano, 2008*).

Firing rate homeostasis is facilitated by the ability of different ion channel combinations to produce equivalent excitability (*Marder, 2011*). The ability of distinct elements to produce the same outcome is known as degeneracy (*Edelman and Gally, 2001*) and has attracted increasing attention in neuroscience (*Marder, 2011*; *Marder and Goaillard, 2006*; *Cropper et al., 2016*; *O'Leary, 2018*; *Ratté and Prescott, 2016*; *Tononi et al., 1999*; *Goaillard and Dufour, 2014*; *Mason et al., 2015*; *Rathour and Narayanan, 2019*). Many aspects of neural function at the genetic (*Klassen et al., 2011*; *Trojanowski et al., 2014*), synaptic (*Mukunda and Narayanan, 2017*; *Anirudhan and Narayanan, 2015*), cellular (*Kim et al., 2017*; *Taylor et al., 2009*; *Drion et al., 2015*; *Ratté et al., 2014b*; *Mittal and Narayanan, 2018*; *Jain and Narayanan, 2020*; *Günay et al., 2008*; *Migliore et al., 2018*), and network (*Grashow et al., 2010*; *Marder and Taylor, 2011*; *Prinz et al., 2004*; *Price and Friston, 2002*; *Onasch and Gjorgjieva, 2020*) levels are now recognized as being degenerate. Ion channel degeneracy facilitates robust homeostatic regulation of neuronal excitability by enabling a disruptive change in one ion channel to be offset by compensatory changes in other ion channels (*O'Leary, 2018*; *Drion et al., 2015*; *Ratté et al., 2014b*; *Rho and Prescott, 2012*; *Swensen and Bean, 2005*; *Zhao and Golowasch, 2012*; *Achard and De Schutter, 2006*; *Bhalla and Bower, 1993*; *Olypher and Calabrese, 2007*).

But to maintain good neural coding, neurons must regulate various aspects of their activity beyond average firing rate (*Stemmler and Koch, 1999*), and must do so while also regulating their energy usage, osmolarity, pH, protein levels, etc. (*Frere and Slutsky, 2018*; *Fisher et al., 2010*; *Balch et al., 2008*). These other cellular properties affect and are affected by neuronal activity, which is to say that properties are not regulated in isolation from one another. And just as each property depends on multiple ion channels, each ion channel can affect multiple properties (*Taylor et al., 2009*; *Günay et al., 2008*). This pleiotropy or functional overlap (*Goaillard and Marder, 2021*) confounds the independent regulation of each property. In response to a perturbation, a compensatory ion channel

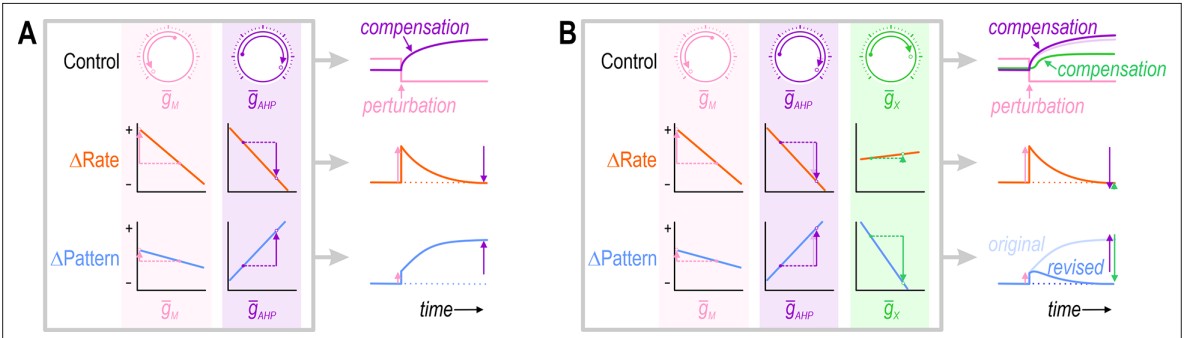

**Figure 1.** Simultaneous regulation of >1 property in a theoretical neuron. (**A**) Challenge: If two ion channels, $g_M$ (pink) and $g_{AHP}$ (purple), both affect firing rate (orange), then the change in firing rate caused by perturbing $g_M$ can be offset by a compensatory change in $g_{AHP}$. However, if $g_M$ and $g_{AHP}$ also affect firing pattern (blue), but not in exactly the same way, then the compensatory change in $g_{AHP}$ that restores firing rate to its target value may exacerbate rather than resolve the change in firing pattern. (**B**) Solution: Adjusting $g_{AHP}$ and at least one additional conductance, $g_x$ (green), that affects each property in a different way than $g_{AHP}$, may enable firing rate and firing pattern to be regulated back to their target values. This suggests that the number of adjustable ion channels ('dials') relative to the number of regulated properties is important. Note that adjusting $g_x$ to restore the firing pattern necessitates a small extra increase in $g_{AHP}$ (compare pale and dark purple); in other words, conductances must be coadjusted.

change that restores one property to its target value may exacerbate rather than mitigate the disturbance of a second property (*Figure 1A*). In theory, both properties could be regulated by adjusting two or more ion channels (*Figure 1B*), but this suggests that the number of properties that can be coregulated is limited by the number of adjustable ion channels, even if the channels are pleiotropic.

To identify the requirements for coregulating many properties, we began with an experiment to confirm that regulating one property by adjusting a single channel is liable to disrupt other properties. We then proceeded with modeling to unravel how multiple channels must be coadjusted to ensure proper regulation of >1 property. We show that the number of adjustable ion channels ($n_{in}$) must exceed the number of regulated properties ($n_{out}$). This is consistent with past work (*Olypher and Prinz, 2010*; *Olypher and Calabrese, 2007*; *Foster et al., 1993*) but the implications for homeostatic regulation have not been explored. To that end, we show that the dimensionality of solution space ($n_{in} - n_{out}$) influences the emergence of ion channel correlations in the presence or absence of noise, and that increased correlations presage regulation failure. We also show that regulating different properties requires that ion channels are coadjusted with different ratios, which necessitates separate master regulators.

## Results

### Adjusting an ion channel to regulate one property risks disrupting other properties – experiments

To test experimentally if regulating one property by adjusting one ion channel is liable to disrupt a second property, we disrupted the firing rate of a CA1 pyramidal neuron by blocking its voltage-gated M-type K$^+$ current ($I_M$) and then we restored firing rate to its baseline value by inserting a virtual calcium-activated AHP-type K$^+$ current ($I_{AHP}$) using dynamic clamp, while also monitoring the firing pattern. Specifically, the neuron was stimulated by injecting irregularly fluctuating (noisy) current under four conditions (*Figure 2A, B*): at baseline (blue), after blocking native $I_M$ (black), and again after introducing virtual $I_M$ (cyan) or virtual $I_{AHP}$ (red). Adding virtual $I_M$ demonstrates that we can replace a native current with an equivalent virtual current; compensation was modeled by inserting a distinct current with functional overlap, namely $I_{AHP}$ (see *Figure 1*). Inserting virtual $I_M$ or virtual $I_{AHP}$ reversed the depolarization (*Figure 2C*), spike amplitude attenuation (*Figure 2D*), and firing rate increase (*Figure 2E*) caused by blocking native $I_M$. Replacing native $I_M$ with virtual $I_M$ did not affect firing pattern, quantified as the coefficient of variation of the interspike interval (CV$_{ISI}$), whereas virtual $I_{AHP}$ reduced CV$_{ISI}$ (*Figure 2F*), often causing the neuron to spike at different times (in response to different stimulus fluctuations) than the neuron with native or virtual $I_M$ (*Figure 2G*). The results confirm predictions from *Figure 1*, namely that compensatory 'upregulation' of $I_{AHP}$ restored firing rate but disrupted firing pattern.

### Few of the ion channel combinations producing the target value for one property also produce the target value for a second property

We proceeded with computational modeling to identify the conditions required to coregulate multiple properties. For our investigation, it was critical to account for all ion channels that are changing. Even for well characterized neurons like pyramidal cells, there is no comprehensive list of which properties are directly regulated and which ion channel are involved in each case. Therefore, we built a simple model neuron in which >1 property can be coregulated by adjusting a small number of ion channels. The relative rates at which ion channel densities are updated using the 'error' in each property differ between properties, meaning ion channels are coadjusted according to a certain ratio to regulate one property (*O'Leary et al., 2014*) but are coadjusted according to a different ratio to regulate a second property, which has some notable implications (see Discussion). We focused on regulation of independent properties like firing rate and energy efficiency per spike – instead of energy consumption rate, for example, which depends on firing rate – but our conclusions do not hinge on which properties are considered. Target values were chosen arbitrarily.

Simulations were conducted in a single-compartment model neuron whose spikes are generated by a fast sodium conductance and a delayed rectifier potassium conductance with fixed densities. A spike-dependent adaptation mechanism, $g_{AHP}$ was included at a fixed density. Densities of all other channels were either systematically varied (and values producing the target output were selected) or

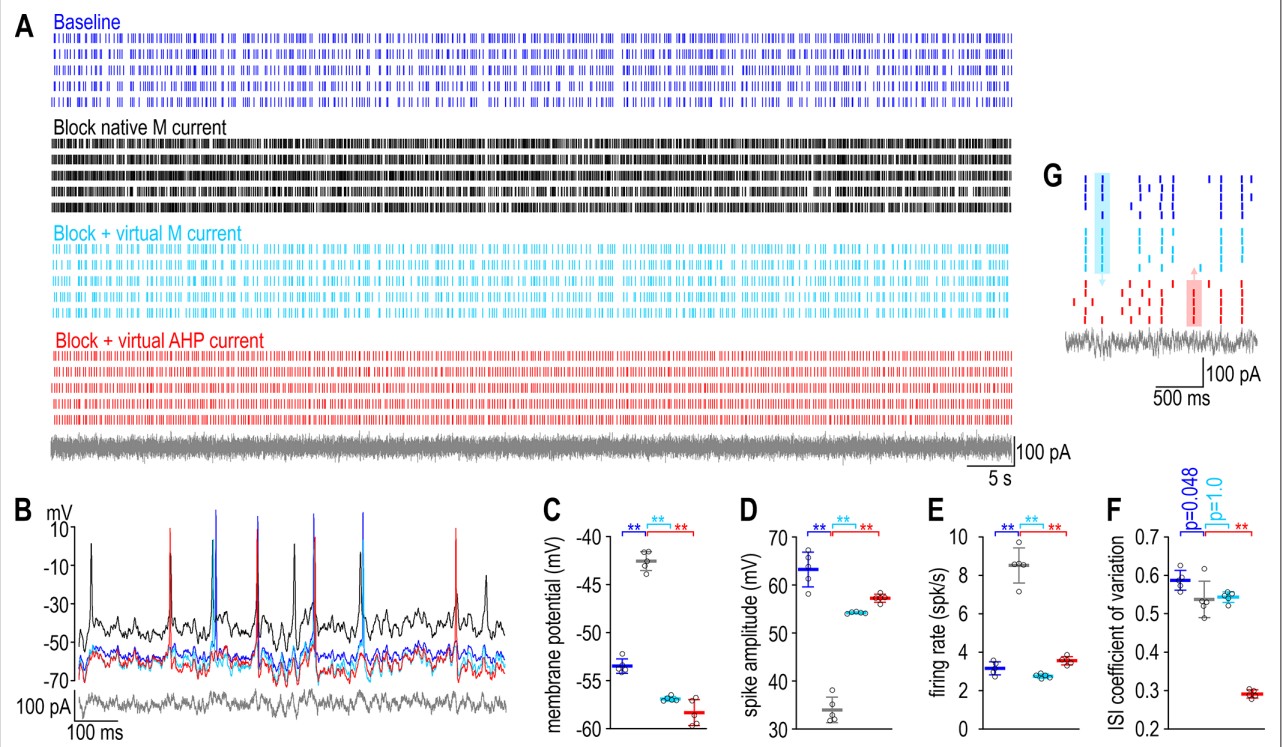

**Figure 2.** A compensatory change that restores firing rate disrupts firing pattern in a CA1 pyramidal neuron. (**A**) Rasters show spiking in a single CA1 pyramidal neuron given the same noisy current injection (gray) on five trials under each of four conditions: baseline (blue), after blocking native $I_M$ with 10 μM XE991 (black), and again after inserting virtual $I_M$ (cyan) or virtual $I_{AHP}$ (red) using dynamic clamp. See Methods for virtual conductance parameters. (**B**) Sample voltage traces under each condition. (**C**) Membrane potential differed across conditions ($F_{3,16}$ = 298.49, p < 0.001, one-way ANOVA); specifically, it was depolarized by blocking $I_M$ ($t$ = 18.71, p < 0.001, Tukey test) but that effect was reversed by inserting virtual $I_M$ ($t$ = 24.56, p < 0.001) or virtual $I_{AHP}$ ($t$ = 27.00, p < 0.001). (**D**) Spike amplitude also differed across conditions ($F_{3,16}$ = 154.04, p < 0.001); specifically, it was attenuated by blocking $I_M$ ($t$ = 20.23, p < 0.001) but that effect was reversed by inserting virtual $I_M$ ($t$ = 13.99, p < 0.001) or virtual $I_{AHP}$ ($t$ = 16.09, p < 0.001). (**E**) Firing rate also differed across conditions ($F_{3,16}$ = 177.61, p < 0.001); specifically, it was increased by blocking $I_M$ ($t$ = 18.38, p < 0.001) but that effect was reversed by inserting virtual $I_M$ ($t$ = 20.85, p < 0.001) or virtual $I_{AHP}$ ($t$ = 16.12, p < 0.001). (**F**) Regularity of spiking, reflected in the coefficient of variation of the interspike interval, also differed across conditions ($F_{3,16}$ = 188.23, p < 0.001), but whereas blocking $I_M$ had a modest effect ($t$ = 2.69, p = 0.048) and inserting virtual $I_M$ had no effect ($t$ = 0.39, p = 1.00), inserting virtual $I_{AHP}$ had a large effect ($t$ = 18.23, p < 0.001). Data are summarized as mean ± standard deviation (SD). Each data point represents a different trial from a single neuron. (**G**) Enlarged view of rasters to highlight spikes that occurred with native or virtual $I_M$ but not with virtual $I_{AHP}$ (blue shading) or vice versa (red shading) to illustrate the change in spike pattern caused by correcting the change in firing rate caused by blocking $I_M$ with a compensatory change in $I_{AHP}$.

The online version of this article includes the following source data for figure 2:

**Source data 1.** Numerical values for experimental data plotted in *Figure 2*.

they were adjusted via negative feedback to produce the target output (see Methods). The former approach, which amounts to a grid search, identifies all density combinations producing a target output (i.e., the solution set). The latter approach finds a subset of those combinations through a homeostatic regulation mechanism. Ion channels with adjustable densities included a sodium conductance ($g_{Na}$) and potassium conductance ($g_K$) that both activate with kinetics similar to the delayed rectifier channel, a slower-activating M-type potassium conductance ($g_M$), and a leak conductance ($g_{leak}$).

We began by testing whether ion channel density combinations that produce the target value for one property also produce a consistent value for other properties. *Figure 3A* shows rheobase for different combinations of $\bar{g}_{Na}$ and $\bar{g}_K$. The contour from point $a$ ($\bar{g}_{Na}$ = 1.54 mS/cm², $\bar{g}_K$ = 0 mS/cm²) to point $b$ ($\bar{g}_{Na}$ = 4 mS/cm², $\bar{g}_K$ = 2.95 mS/cm²) represents all combinations (i.e., the solution set) yielding a rheobase of 30 μA/cm². Despite yielding the same rheobase, density combinations along the $a$–$b$ contour did not yield the same minimum sustainable firing rate ($f_{min}$) (*Figure 3B*). The value of $f_{min}$ reflects spike initiation dynamics: $f_{min} \gg 0$ spk/s is consistent with class 2 excitability (*Hodgkin, 1948*)

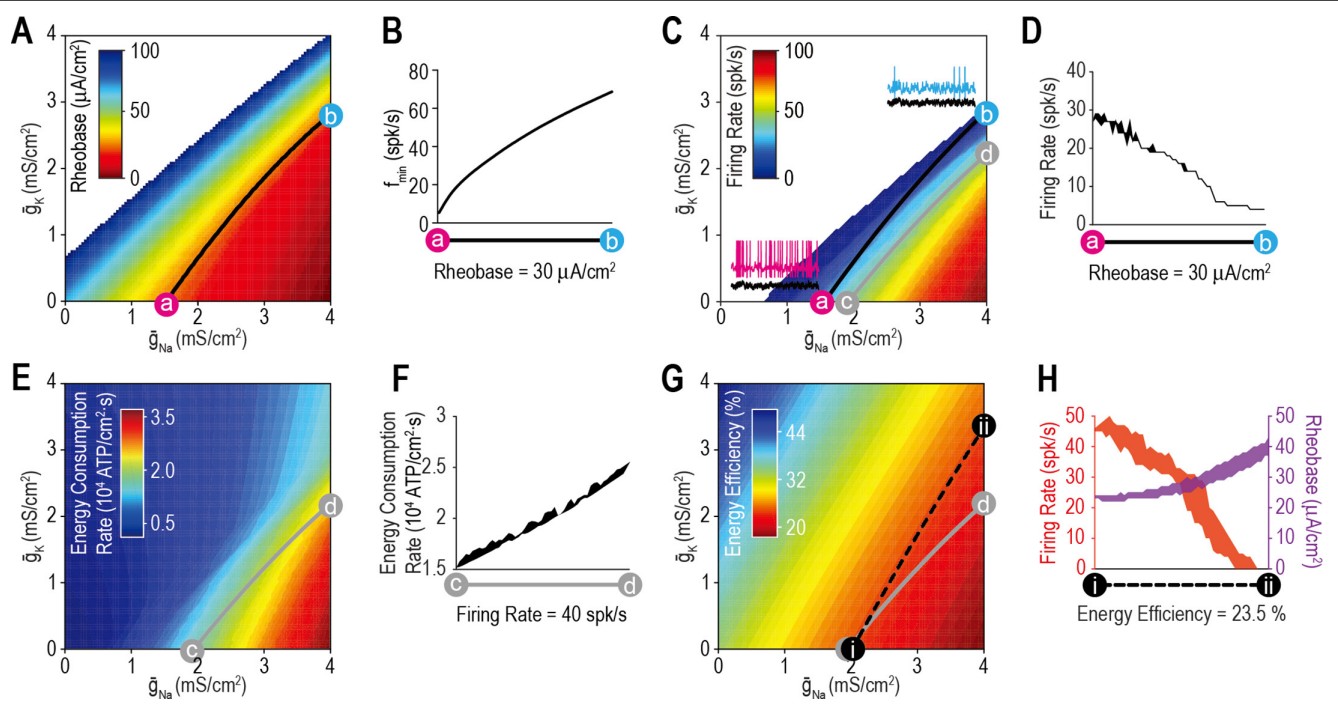

**Figure 3.** Most ion channel combinations producing the target value for one property produce inconsistent values for other properties. (**A**) Color shows the minimum $I_{stim}$ required to evoke repetitive spiking (rheobase) for different combinations of $\bar{g}_{Na}$ and $\bar{g}_K$. Contour linking *a* and *b* highlights density combinations yielding a rheobase of 30 µA/cm². (**B**) Minimum sustainable firing rate ($f_{min}$) varied along the iso-rheobase contour (see ***Figure 3—figure supplement 1***). (**C**) Color shows firing rate evoked by noisy $I_{stim}$ ($\tau_{stim}$ = 5ms, $\sigma_{stim}$ = 10 µA/cm², $\mu_{stim}$ = 40 µA/cm²) for different combinations of $\bar{g}_{Na}$ and $\bar{g}_K$. Adding spike-dependent and -independent forms of adaptation ($\bar{g}_{AHP}$ = 1.75 mS/cm² and $\bar{g}_M$ = 0.5 mS/cm²) broadened the dynamic range. Gray curve shows density combinations yielding a firing rate of 40 spk/s (i.e., an iso-firing rate contour; shown in red in ***Figures 4–6***). Insets show sample responses to equivalent noisy stimulation. (**D**) Firing rate varied along the iso-rheobase contour from panel A. (**E**) Color shows energy consumption rate for different combinations of $\bar{g}_{Na}$ and $\bar{g}_K$ based on firing rates shown in panel C. Iso-firing rate contour *c–d* (gray line in panel C) does not align with energy contours. ATP consumed by the Na⁺/K⁺ pump was calculated from the total Na⁺ influx and K⁺ efflux determined from the corresponding currents. (**F**) Energy consumption rate varied along the iso-firing rate contour. (**G**) Color shows energy efficiency per spike for different combinations of $\bar{g}_{Na}$ and $\bar{g}_K$. Energy efficiency was calculated as the capacitive minimum, $C\Delta V$, divided by total Na⁺ influx, where $C$ is capacitance and $\Delta V$ is spike amplitude (see ***Figure 3—figure supplement 2***). Density combinations along contour *i–ii* (dashed line) yield energy efficiency of 23.5% (shown in green in ***Figure 5***). (**H**) Both rheobase and firing rate varied along the iso-energy efficiency contour.

The online version of this article includes the following figure supplement(s) for figure 3:

**Figure supplement 1.** Ion channel combinations yielding equivalent rheobase produce different responses to noisy stimulation, reflecting fundamental differences in spike initiation dynamics and operating mode.

**Figure supplement 2.** Overlap between Na⁺ and K⁺ currents dictates energy efficiency.

and operation as a coincidence detector (***Ratté et al., 2014a***), whereas $f_{min} \approx 0$ spk/s is consistent with class 1 excitability and operation as an integrator, with several consequences (***Figure 3—figure supplement 1***). After adding $g_{AHP}$ to expand the stimulus range over which fluctuation-driven spikes occur, we plotted the firing rate driven by irregularly fluctuating (noisy) $I_{stim}$ for different combinations of $\bar{g}_{Na}$ and $\bar{g}_K$ (***Figure 3C***). Density combinations yielding the same rheobase did not yield equivalent stimulation-evoked firing rates (***Figure 3D***).

Having demonstrated that ion channel density combinations yielding the target value for one aspect of excitability (e.g., rheobase) yield differing values for other aspects of excitability (e.g., $f_{min}$ or firing rate), we predicted that the same lack of generalization would extend to other cellular properties (e.g., energy efficiency). Spikes are energetically costly (***Attwell and Laughlin, 2001***) but vary in their energy efficiency based on the temporal overlap in sodium and potassium channel activation (***Sengupta et al., 2010***; ***Figure 3—figure supplement 2***). Using responses reported in ***Figure 3C***, we measured energy consumption rate for combinations of $\bar{g}_{Na}$ and $\bar{g}_K$ (***Figure 3E***). Energy consumption rate increased with firing rate, as expected, but did not increase equivalently across all density combinations, as evident from the variation in energy consumption rate along the iso-firing rate

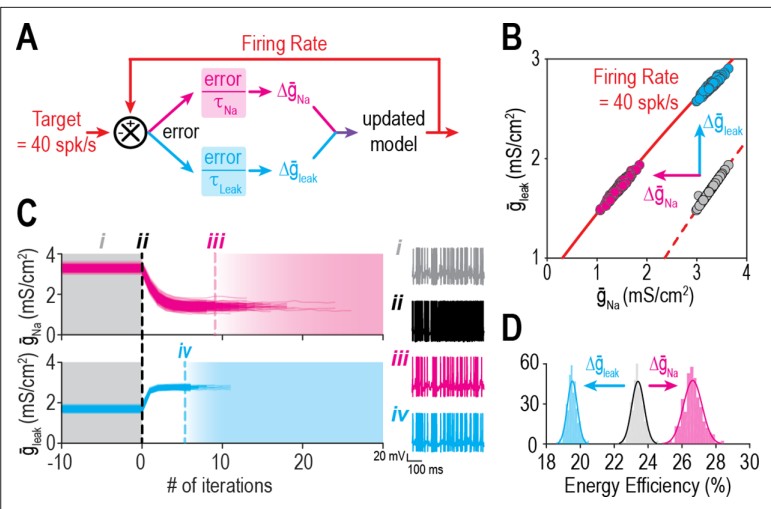

**Figure 4.** Ion channel changes mediating the same effect on firing rate can oppositely affect energy efficiency. (**A**) Schematic shows how a difference in firing rate from its target value creates an error that is reduced by updating $\bar{g}_{Na}$ or $\bar{g}_{leak}$. (**B**) Iso-firing rate contours for 40 spk/s are shown for different combinations of $\bar{g}_{Na}$ and $\bar{g}_{leak}$ with $\bar{g}_{K}$ at baseline (2 mS/cm², dashed red curve) and after $\bar{g}_{K}$ was 'knocked out' (0 mS/cm², solid red curve). When $\bar{g}_{K}$ was abruptly reduced, starting models (gray dots) spiked rapidly (~93 spk/s) before firing rate was regulated back to its target value by compensatory changes in either $\bar{g}_{Na}$ (pink) or $\bar{g}_{leak}$ (cyan). Models evolved in different directions and settled at different positions along the solid curve. (**C**) Trajectories show evolution of $\bar{g}_{Na}$ and $\bar{g}_{leak}$. Trajectories are terminated once target firing rate is reached. Sample traces show responses before (gray) and immediately after (black) $\bar{g}_{K}$ was reduced, and again after compensatory changes in $\bar{g}_{Na}$ (pink) or $\bar{g}_{leak}$ (cyan). (**D**) Distributions of energy efficiency are shown before (gray) and after firing rate regulation via control of $\bar{g}_{Na}$ (pink) or $\bar{g}_{leak}$ (cyan).

contour (*Figure 3F*). This is due to differences in energy efficiency (*Figure 3G*) determined as the energy consumed per spike relative to the theoretical minimum (see Methods). Density combinations yielding equally efficient spikes yielded different stimulation-evoked firing rates and rheobase values (*Figure 3H*). One might presume that spikes are produced as efficiently as possible rather than being regulated to a specific value, but energy efficiency decreases propagation safety factor (*Al-Basha and Prescott, 2019*) and a target value likely emerges by balancing these competing interests (*Speakman et al., 2011*).

## Adjusting an ion channel to regulate one property risks disrupting other properties – simulations

Like the experiment in *Figure 2*, we tested if restoring firing rate to its target value via compensatory changes in either of two ion channels (*Figure 4A*) after perturbing a third channel would disrupt energy efficiency in our model neuron. Gray circles on *Figure 4B* show randomly chosen combinations of $\bar{g}_{Na}$ and $\bar{g}_{leak}$ that yield a firing rate of 40 spk/s when $\bar{g}_{K}$ = 2 mS/cm². When $\bar{g}_{K}$ was 'blocked' (abruptly reset to 0 mS/cm²), firing rate jumped to ~93 spk/s before being restored to 40 spk/s via feedback control of either $\bar{g}_{Na}$ (pink) or $\bar{g}_{leak}$ (cyan) (*Figure 4C*). As $\bar{g}_{Na}$ or $\bar{g}_{leak}$ converged on new, compensated densities, firing rate returned to its target value but energy efficiency was affected in opposite ways (*Figure 4D*). If energy efficiency is also regulated, then $\bar{g}_{Na}$ and $\bar{g}_{leak}$ must be coadjusted in a way that restores firing rate without disrupting energy efficiency.

## Geometrical explanation for the relationship between single- and multioutput solutions

Next, we sought a geometrical explanation for how to coadjust >1 ion channel to coregulate >1 property. The top panel of *Figure 5A* shows all combinations of $\bar{g}_{Na}$ and $\bar{g}_{K}$ ($n_{in}$ = 2) yielding the target value for firing rate (red) *or* energy efficiency (green) ($n_{out}$ = 1). Like in *Figures 3 and 4*, the solution set for each property corresponds to a curve. The multioutput solution set for firing rate *and* energy efficiency ($n_{out}$ = 2) corresponds to where the two single-output solution sets intersect, which occurs at a point – only one density combination yields the target values for both properties, meaning the

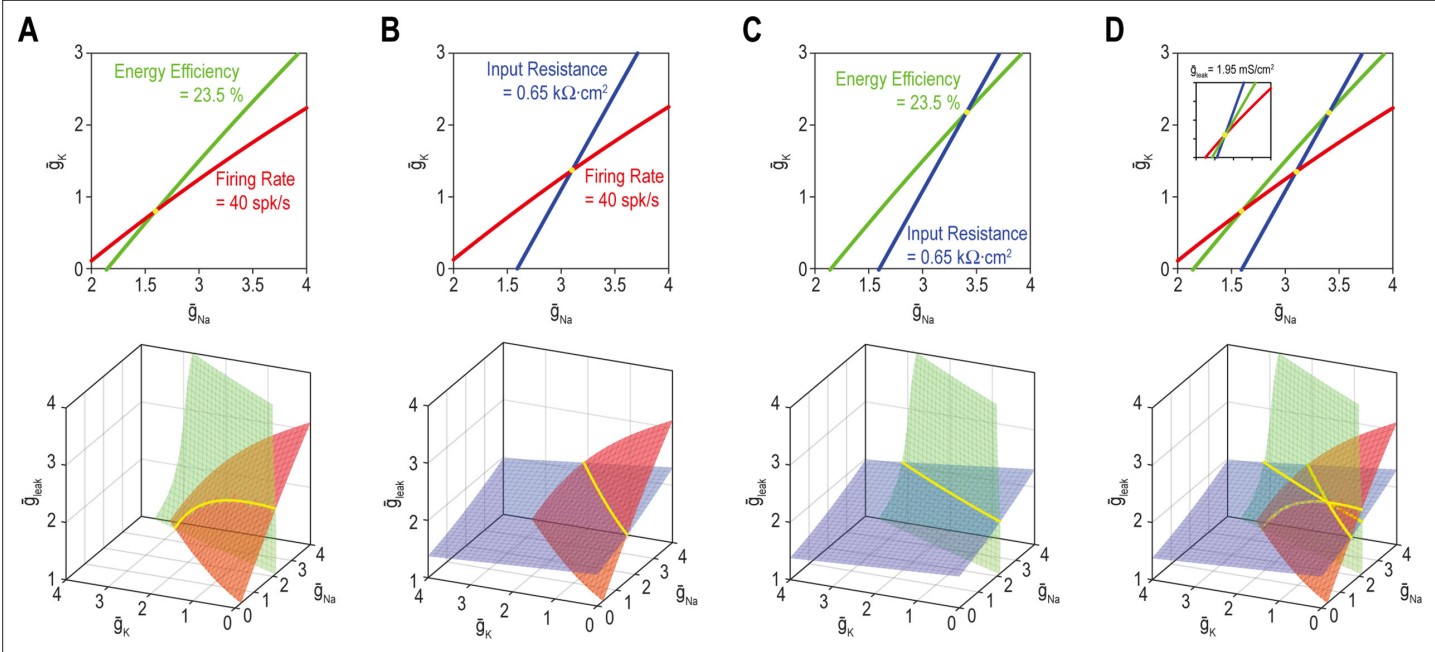

**Figure 5.** A degenerate solution for *n* properties requires at least *n* + 1 adjustable ion channels. Curves in top panels depict single-output solutions sets based on control of $\bar{g}_{Na}$ and $\bar{g}_K$ ($n_{in}$ = 2); $\bar{g}_{leak}$ = 2 mS/cm². Surfaces in bottom panels depict single-output solution sets based on control of $\bar{g}_{Na}$, $\bar{g}_K$, and $\bar{g}_{leak}$ ($n_{in}$ = 3). Intersection (yellow) of single-output solutions at a point constitutes a unique multioutput solution, whereas intersection along a curve (or higher-dimensional manifold) constitutes a degenerate multioutput solution. (**A**) Curves for firing rate (40 spk/s) and energy efficiency (23.5%) intersect at a point whereas the corresponding surfaces intersect along a curve. Solutions for firing rate and input resistance (0.65 kΩ cm²) (**B**) and for energy efficiency and input resistance (**C**) follow the same pattern as in panel A. (**D**) For $n_{in}$ = 2 (top), curves for firing rate, energy efficiency and input resistance do not intersect at a common point unless $\bar{g}_{leak}$ is reset to 1.95 mS/cm² (inset). For $n_{in}$ = 3 (bottom), the three surfaces intersect at the same point as in the inset. See *Figure 5—figure supplement 1* for the effects of tolerance on solution sets.

The online version of this article includes the following figure supplement(s) for figure 5:

**Figure supplement 1.** Increasing tolerance does not increase the dimensionality of multioutput solution sets the same way as increasing $n_{in}$.

multioutput solution is unique. But if another conductance like $g_{leak}$ is adjustable ($n_{in}$ = 3), the curves in 2D parameter space (top panel) transform into surfaces in 3D parameter space (bottom panel) and those surfaces intersect along a curve – many density combinations yield the target values for both properties, meaning the multioutput solution is degenerate. The same patterns are evident for firing rate *and* input resistance (*Figure 5B*) or energy efficiency *and* input resistance (*Figure 5C*). *Figure 5D* shows that if all three properties – firing rate, energy efficiency *and* input resistance – are regulated ($n_{out}$ = 3), the multioutput solution set is empty for $n_{in}$ = 2 (i.e., the three curves do not intersect at a common point [top panel] unless $\bar{g}_{leak}$ is reset to 1.95 mS/cm² [inset]) and is unique for $n_{in}$ = 3 (bottom panel). From this, we conclude that robust regulation of *n* properties requires >*n* adjustable ion channels even if each ion channel contributes to regulation of >1 property. This is like a system of linear equations, which is said to be *underdetermined* if unknowns (inputs) outnumber equations (outputs) (see Discussion).

In all simulations involving homeostatic regulation, properties were regulated to within certain bounds of the target rather than to a precise value, despite how solution sets are depicted. When tolerances are depicted, single-output solution sets correspond to thin strips (rather than curves) or shallow volumes (rather than surfaces) in 2D and 3D parameter space, respectively (*Figure 5—figure supplement 1*). The width, depth, etc. are proportional to the tolerance. How does this impact multioutput solutions? *Thin* 2D strips intersect at a small patch (unlike curves intersecting at a point) but that patch in 2D parameter space (*Figure 5—figure supplement 1A*) is unlike the long 1D curve formed by *broad* 2D surfaces intersecting in 3D parameter space (see bottom panels of *Figure 5*). Likewise, *shallow* 3D volumes intersect at a narrow tube (unlike surfaces intersecting at a curve) but that tube in 3D parameter space (see *Figure 5—figure supplement 1B*) is unlike the broad 2D surface

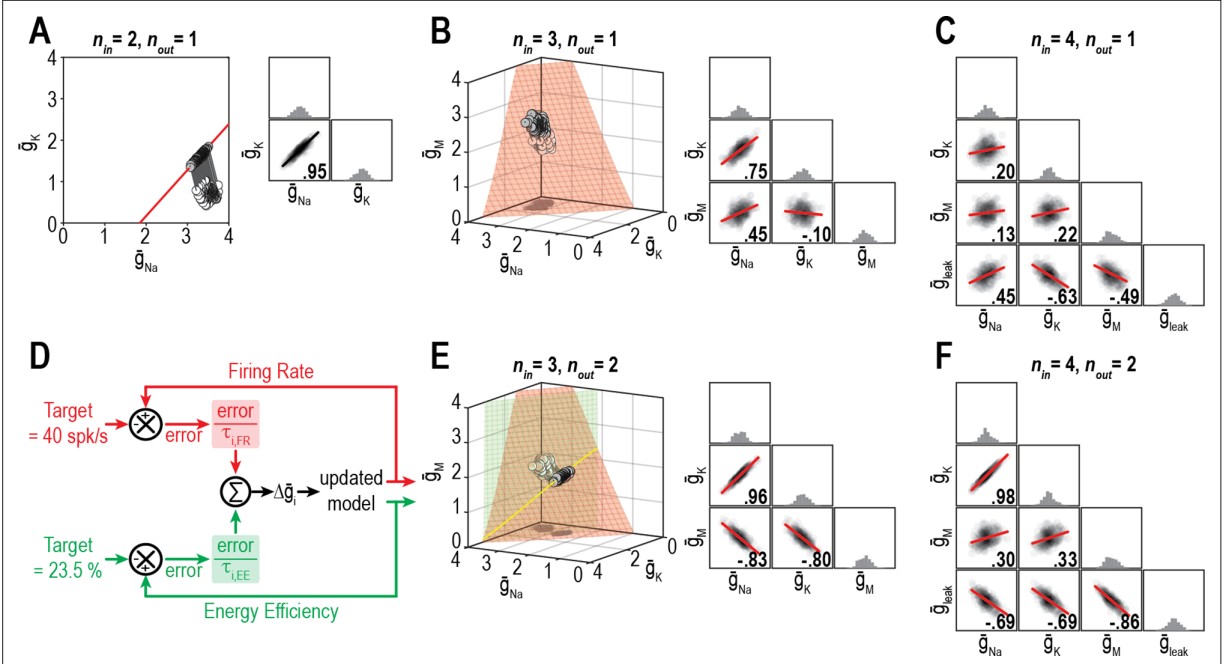

**Figure 6.** Dimensionality of solution space affects ion channel correlations. (**A**) Starting from a normally distributed cluster of $\bar{g}_{Na}$ and $\bar{g}_K$ (white dots) yielding an average firing rate of 73 spk/s, $\bar{g}_{Na}$ and $\bar{g}_K$ ($n_{in}$ = 2) were homeostatically adjusted to regulate firing rate ($n_{out}$ = 1) to its target value of 40 spk/s. Gray lines show trajectories. Because solutions (gray dots) converge on a curve, the pairwise correlation between $\bar{g}_{Na}$ and $\bar{g}_K$ is predictably strong. Scatter plots show solutions centered on the mean and normalized by the standard deviation (z-scores). Correlation coefficient (R) is shown in the bottom right corner of each scatter plot. (**B**) Same as panel A ($n_{out}$ = 1) but via control of $\bar{g}_{Na}$, $\bar{g}_K$, and $\bar{g}_M$ ($n_{in}$ = 3). Homeostatically found solutions converge on a surface and ion channel correlations are thus weaker. (**C**) If $\bar{g}_{leak}$ is also controlled ($n_{in}$ = 4), solutions converge on a hard-to-visualize volume (not shown) and pairwise correlations are further weakened. (**D**) Schematic shows how errors for two regulated properties are combined: the error for each property is calculated separately and is scaled by its respective control rate $\tau$ to calculate updates, and all updates for a given ion channel (i.e., originating from each error signal) are summed. (**E**) Same as panel B ($n_{in}$ = 3), but for regulation of firing rate and energy efficiency ($n_{out}$ = 2). Homeostatically found solutions once again converge on a curve (yellow), which now corresponds to the intersection of two surfaces; ion channel correlations are thus strong, like in panel A. (**F**) If $n_{in}$ is increased to four while $n_{out}$ remains at 2, solutions converge on a surface (not shown) and ion channel correlations weaken.

formed by *deep* 3D volumes intersecting in 4D parameter space. In other words, increasing tolerance does not increase solution space dimensionality the same way as increasing $n_{in}$.

The need for $\geq n$ adjustable channels to regulate $n$ properties has been shown before (*Olypher and Prinz, 2010*; *Olypher and Calabrese, 2007*; *Foster et al., 1993*) and may be obvious to some for mathematical reasons, but the relationship has some important implications (e.g., for parameter estimation *Sarkar and Sobie, 2010*). The implications for homeostatic regulation have not been thoroughly explored, thus prompting the next steps of our study.

## The dimensionality of solution space affects ion channel correlations

We predicted that the dimensionality of solution space – point (0D), curve (1D), surface (2D), volume (3D), etc. – affects ion channel correlations by limiting the degrees of freedom. To explore this, we measured ion channel correlations within single- and multioutput solution sets found through homeostatic regulation. *Figure 6A* shows all combinations of $\bar{g}_{Na}$ and $\bar{g}_K$ ($n_{in}$ = 2) producing the target firing rate ($n_{out}$ = 1). Correlation between $\bar{g}_{Na}$ and $\bar{g}_K$ is high because homeostatically determined solutions are constrained to fall along a curve; under those conditions, variation in one channel is offset solely by variation in the other channel, and all covariance is thus captured in a single pairwise relationship. If $\bar{g}_M$ is also allowed to vary ($n_{in}$ = 3), homeostatically determined solutions spread across a surface and pairwise correlations become predictably weaker (*Figure 6B*) since variation in one channel can be offset by variation in two other channels, and covariance is diluted across >1 pairwise relationship. If additional channels are allowed to vary ($n_{in} \geq 4$), solutions distribute over higher-dimensional manifolds and pairwise correlations further weaken (*Figure 6C*). However, if firing rate *and* energy efficiency are

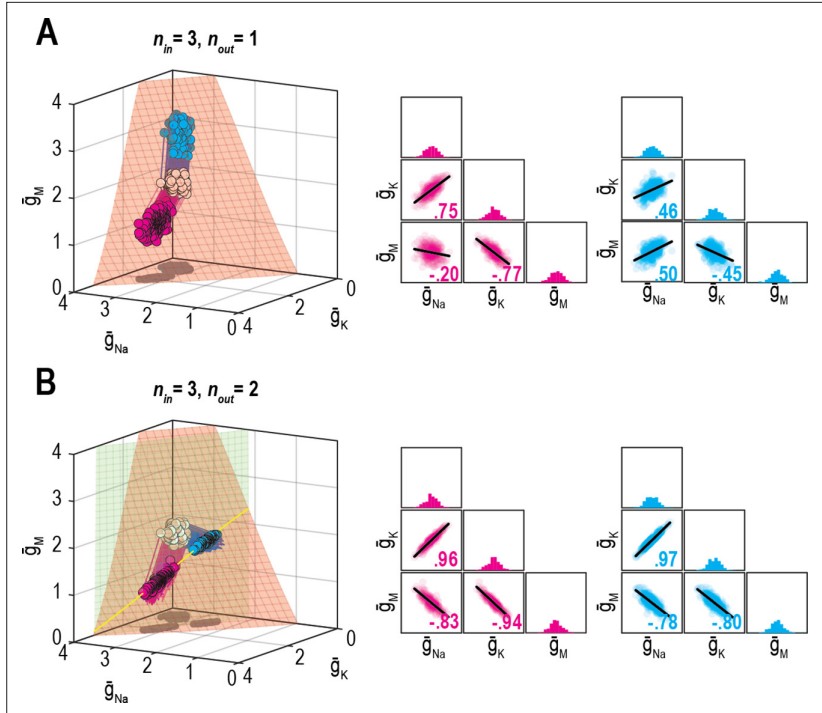

**Figure 7.** Effect of relative regulation rates on ion channel correlations depends on the dimensionality of solution space. (**A**) Homeostatic regulation of firing rate via control of $\bar{g}_{Na}$, $\bar{g}_K$, and $\bar{g}_M$. Same as **Figure 6B**, but for two new sets of regulation rates (see **Supplementary file 1**). Solutions found for each set of rates (pink and cyan) approached the surface from different angles and converged on the surface with different patterns, thus producing distinct ion channel correlations, consistent with **O'Leary et al., 2013**. (**B**) Same as panel A ($n_{in}$ = 3), but for homeostatic regulation of firing rate and energy efficiency ($n_{out}$ = 2). Solutions converge on a curve (yellow), giving rise to virtually identical ion channel correlations regardless of regulation rates.

both regulated ($n_{out}$ = 2; **Figure 6D**), homeostatically determined solutions are once again constrained to fall along a curve when $n_{in}$ = 3, and pairwise correlations strengthen (**Figure 6E**). Increasing $n_{in}$ to four while keeping $n_{out}$ at two caused correlations to weaken (**Figure 6F**). These results confirm the predicted impact of solution space dimensionality on ion channel correlations.

*O'Leary et al., 2013* demonstrated how the relative rates at which different ion channel densities are controlled impact their correlation. This is reproduced in **Figure 7A**, where, from the same initial conditions (density combinations), homeostatic control with different relative rates (shown in pink and cyan) produces solutions with different correlations. Relative regulation rates can affect not only the strength of pairwise correlations, but also the sign (compare correlation between $\bar{g}_{Na}$ and $\bar{g}_M$). However, if firing rate *and* energy efficiency are both homeostatically regulated, correlations strengthen (consistent with results from **Figure 6**) and become independent of the relative regulation rates because solutions are limited to a lower-dimensional solution space (**Figure 7B**). Recall that dimensionality of the multioutput solution set corresponds to $n_{in} - n_{out}$. For relative regulation rates to influence ion channel correlations, homeostatically determined solutions must fall on a solution manifold with dimensionality >1, but not >>1 lest pairwise correlations be diluted.

However, correlations produced through the homeostatic regulation mechanism described by *O'Leary et al., 2013*, which is essentially the same mechanism used here, were recently shown to be sensitive to noise (*Franci et al., 2020*). This occurs because this regulation mechanism brings conductance densities to the solution manifold (by minimizing the error signal) but cannot control how solutions drift on the manifold (since the error signal is 0 everywhere on the manifold). Solutions do not drift in the absence of noise but, of course, noise is ubiquitous in biological systems and its effects must be considered. Accordingly, we reran simulations from **Figures 6 and 7** with noise added to the conductance densities. After a few hundred noise-update iterations, solutions spread across the available solution space, which corresponds to a surface when only firing rate is regulated (**Figure 8A**, top).

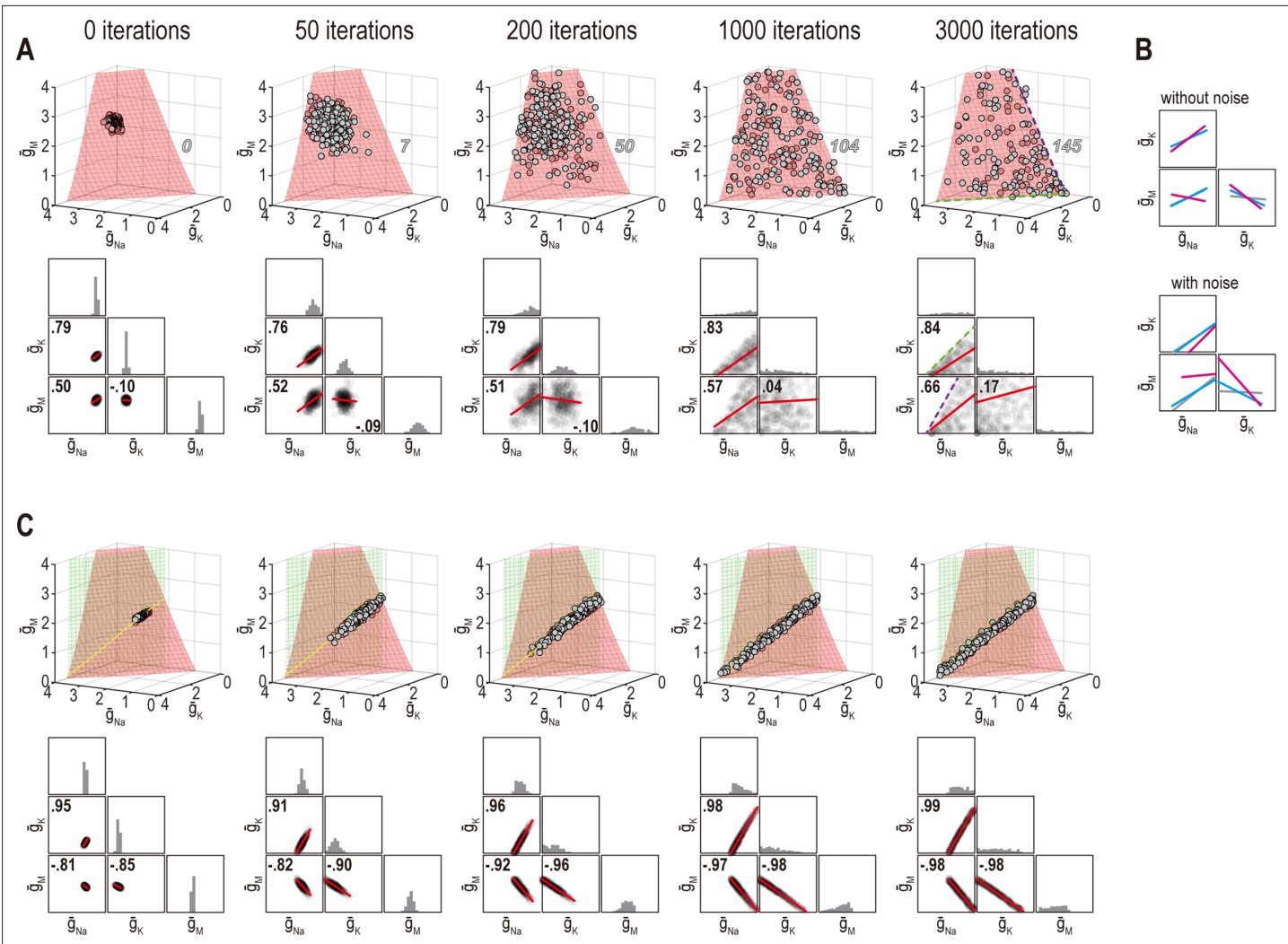

**Figure 8.** Noise can affect ion channel correlations depending on the dimensionality of solution space. (**A**) Starting from solution sets in *Figure 6B*, $\bar{g}_{Na}$, $\bar{g}_K$, and $\bar{g}_M$ had noise added to them and were then updated by homeostatic regulation (using regulation rates from *Figure 6*) to correct the noise-induced disruption of firing rate. Conductance density combinations are depicted before and after 50, 200, 1000, or 3000 noise-update iterations (left to right). Noise caused solutions to spread across the surface (top). Some solutions (number in italics) drifted beyond the illustrated region but this is prevented by imposing an upper bound (*Figure 8—figure supplement 1A*). Ion channel correlations reflect the available solution space; for example, changes in $\bar{g}_{Na}$ and $\bar{g}_K$ are limited by $\bar{g}_M$ remaining positive (dashed green line), while changes in $\bar{g}_{Na}$ and $\bar{g}_M$ are limited by $\bar{g}_K$ remaining positive (dashed purple line). How solutions distribute within that space depends on regulation rates, and also affects correlations (*Figure 8—figure supplement 1B, C*). Conductance density distributions and pairwise correlations (bottom) were not centered on the mean or normalized by standard deviation (unlike in other figures) in order to visualize how distributions evolve over iterations. (**B**) Comparison of regression lines from *Figure 6* (gray) and 7 (pink, cyan) without noise (top) and with noise (bottom). Correlations are affected by regulation rates in both conditions, but not in the same way. (**C**) Same as panel A but for regulation of firing rate *and* energy efficiency. Solutions spread along the intersection of the two surfaces, but the spread is bounded by one or another conductance density reaching 0 mS/cm². Correlations slightly increase under noisy conditions (compare with *Figure 6*).

The online version of this article includes the following figure supplement(s) for figure 8:

**Figure supplement 1.** Correlations depend on the solution manifold's shape and how solutions distribute across it.

The distribution of solutions can produce correlations (*Figure 8A*, bottom). Lower and upper bounds on conductance densities shape the solution space and influence correlations (*Figure 8—figure supplement 1A*). Correlations also depend on how solutions distribute across the solution space, which depends on regulation rates (*Figure 8—figure supplement 1B, C*), which means correlations can still depend on regulation rates under noisy conditions, but correlations are not the same as in the absence of noise (*Figure 8B*). When firing rate and energy efficiency were regulated, noise caused solutions to spread across the intersection (*Figure 8C*) but correlations were relatively unaffected

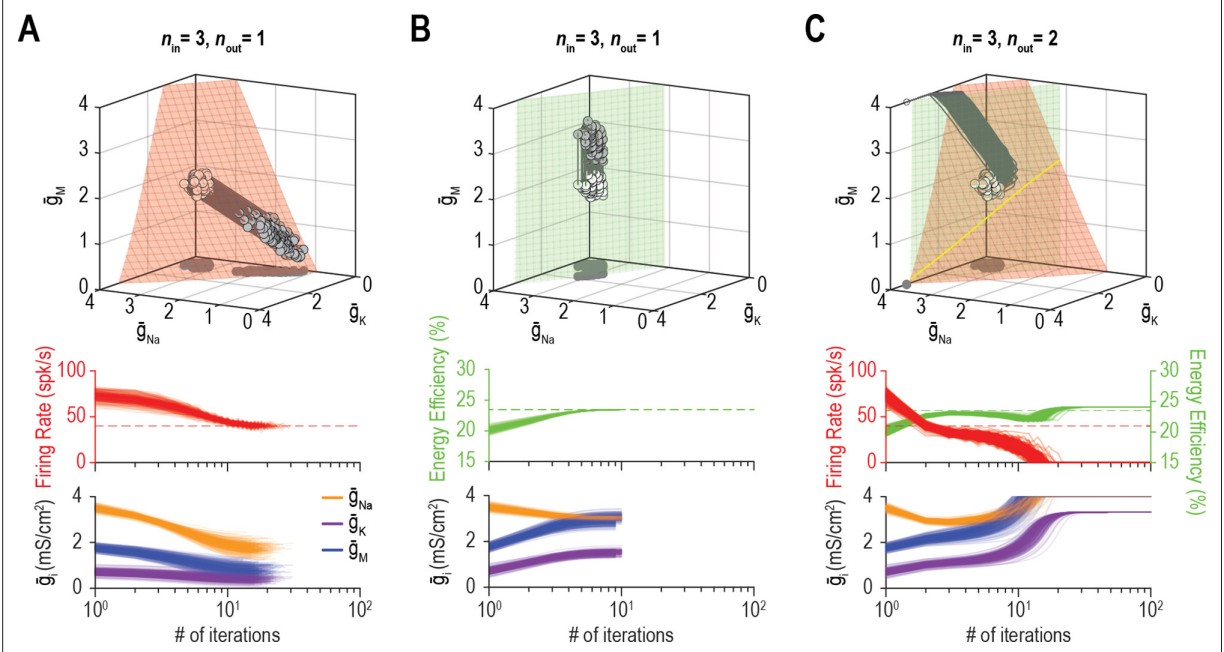

**Figure 9.** Low-dimensional solutions can be hard for homeostatic regulation to 'find'. (**A**) Homeostatic regulation of firing rate ($n_{out}$ = 1) via control of $\bar{g}_{Na}$, $\bar{g}_K$, and $\bar{g}_M$ ($n_{in}$ = 3), like in *Figures 6B and 7A* but using a different set of regulation rates (see *Supplementary file 1*). Solutions converged onto the iso-firing rate surface (top panel) and firing rate was regulated to its target value in <30 iterations (bottom panel). (**B**) Same as A ($n_{in}$ = 3) but for homeostatic regulation of energy efficiency ($n_{out}$ = 1). Solutions converged on the iso-energy efficiency surface (top panel) and energy efficiency was regulated to its target value in ~10 iterations (bottom panel). (**C**) Homeostatic regulation of firing rate and energy efficiency ($n_{out}$ = 2) via control of the same ion channels ($n_{in}$ = 3) using the same relative rates and initial values as in panels A and B. Neither firing rate (red trajectories) nor energy efficiency (green trajectories) reached its target value. Conductance densities were capped at 4 mS/cm² but this does not account for trajectories not reaching their target. Noise was not included in these simulations.

since they are limited by the low dimensionality of this solution space. Spread was bounded by one or another conductance density reaching 0 mS/cm². *Franci et al., 2020* proposed another regulatory scheme in which an attractive subspace emerges through cooperative molecular interactions; correlations created by fluctuations along that subspace are robust to noise. That scheme should preclude correlations from arising as in *Figure 8A* because solutions are prevented from spreading across the solution space; conversely, a low-dimensional solution space like in *Figure 8C* may prevent correlations unless the attractive subspace arising from molecular interactions aligns with the solution space. Further work is required to explore these regulatory mechanisms and how they interact.

## The dimensionality of solution space affects the success of homeostatic regulation

Beyond affecting the ion channel correlations that emerge through homeostatic regulation, we predicted that the dimensionality of solution space affects whether multiple properties can be successfully regulated to their target values. *Figure 9A* shows an example in which coadjusting $\bar{g}_{Na}$, $\bar{g}_K$, and $\bar{g}_M$ successfully regulates firing rate. *Figure 9B* shows successful regulation of energy efficiency by coadjusting the same three channels. Using the same initial conditions and relative regulation rates as above, the system failed to coregulate firing rate *and* energy efficiency (*Figure 9C*). Notably, coordinated regulation of both properties was achieved using other relative regulation rates (see *Figures 6C and 7B*) or using the same relative rates but starting from different initial conditions (not shown), suggesting that low-dimensional solutions are less accessible (i.e., a smaller set of relative regulation rates succeed in finding the solution space). This may be a challenge for the Franci et al. regulation model if molecular interactions create an attractive subspace that does not align with a low-dimensional solution space.

*Figure 10* shows additional examples of regulating firing rate and energy efficiency by coadjusting $\bar{g}_{Na}$, $\bar{g}_K$, and $\bar{g}_M$. In these simulations, firing rate is regulated to a precise target value whereas energy

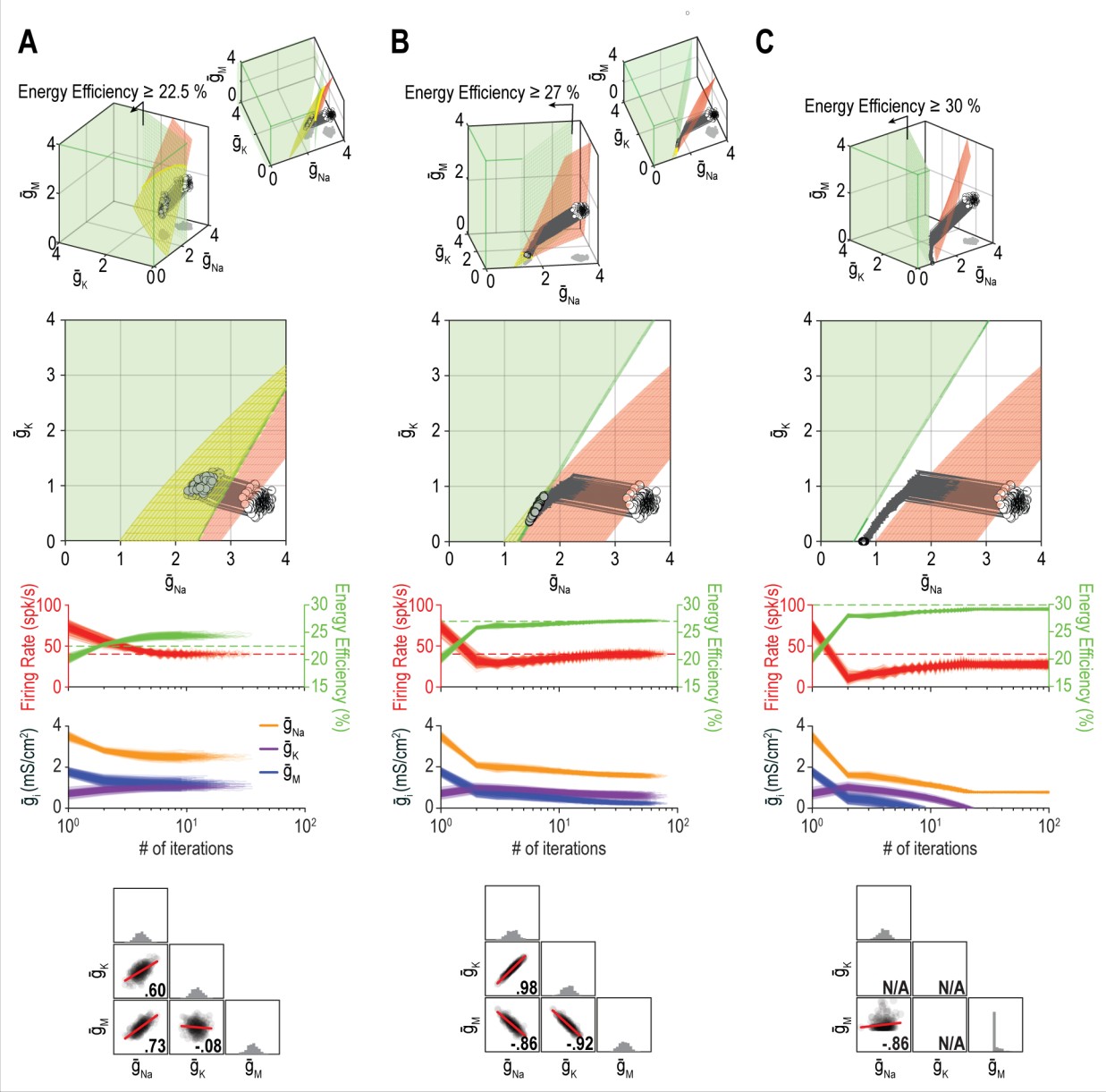

**Figure 10.** The outcome of homeostatic regulation depends on if and how single-output solution sets intersect. Homeostatic regulation of firing rate and energy efficiency ($n_{out}$ = 2) via control of $\bar{g}_{Na}$, $\bar{g}_K$, and $\bar{g}_M$ ($n_{in}$ = 3). For these simulations, energy efficiency was maintained above a lower bound rather than being regulated to a specific target value; accordingly, the single-output solution set for energy efficiency corresponds to a volume (green) rather than a surface. (**A**) For energy efficiency ≥22%, homeostatically determined solutions converge on the iso-firing rate surface (red) in a region sitting within the green volume (top panel). The rate of convergence and resulting ion channel correlations are shown in the middle and bottom panels, respectively. (**B**) For energy efficiency ≥27%, solutions initially converge on the red surface in a region outside the green volume, but trajectories then bend and proceed across the red surface until that surface reaches the green volume. Because solutions converge on a curve, ion channel correlations are stronger than in A, where solutions distributed across a surface. (**C**) For energy efficiency ≥30%, the red surface and green volume do not intersect. Consequently, solutions settle between the two single-output solution sets (top panel) without either property reaching its target (middle panel). The outcome represents the balance achieved by the opposing pull of control mechanisms regulating different properties, and depends entirely on $\bar{g}_{Na}$ since $\bar{g}_K$ and $\bar{g}_M$ cannot become negative (bottom panel). Noise was not included in these simulations.

efficiency is maintained above a lower bound; the single-output solution set for energy efficiency thus corresponds to a volume rather than a surface for $n_{in}$ = 3. For energy efficiency ≥22%, homeostatically determined solutions converge on the iso-firing rate surface without regulation of energy efficiency having much effect (*Figure 10A*). For energy efficiency ≥27%, solutions initially converge onto part of the iso-firing rate surface that sits outside the targeted energy efficiency volume, but without energy

efficiency requirements being met, solutions then move across the surface until reaching the intersection with the iso-energy efficiency volume (*Figure 10B*). By converging on the intersection, which is a curve, ion channel correlations are stronger than in *Figure 10A*. In other words, ion channel correlations increase as single-output solution sets start to disconnect, meaning increased correlations presage regulation failure; in general, this is consistent with the impact of increasing constraints (see *Figure 8*), which go hand in hand with decreasing degrees of freedom. For energy efficiency ≥30%, the iso-firing rate surface and targeted energy efficiency volume do not intersect and solutions thus proceed to a point between the two single-output solution sets constrained by $\bar{g}_K$ and $\bar{g}_M$ reaching 0 mS/cm$^2$ (*Figure 10C*). In this last example, neither property is optimally regulated but the outcome is a reasonable compromise. Regulation might have failed outright if, in the absence of an upper bound, ion channel densities increased (wound-up) without properties ever reaching their target values. This highlights that coregulation of multiple properties might fail or find suboptimal solutions not because single-output solutions do not exist, but because even large single-output solution sets might not intersect, meaning no multioutput solution exists.

## Discussion

This study has identified conditions required for a neuron to coregulate >1 property. Being able to produce the same output using diverse ion channel combinations ensures that single-output solution sets are large. This is critical because of the many ion channel combinations that produce the desired output for one property, few also produce the desired output for other properties (*Figure 3*), which means ion channel adjustments that regulate one property are liable to disrupt other properties due to ion channel pleiotropy (*Figures 2 and 4*). Indeed, the multioutput solution set corresponds to the intersection between single-output solution sets and the dimensionality of the multioutput solution set corresponds to the difference between the number of adjustable ion channels ($n_{in}$) and the number of regulated outputs ($n_{out}$) (*Figure 5*). Coregulation of *n* properties requires at least *n* adjustable ion channels for a unique solution, and at least *n* + 1 channels for a degenerate solution. This constraint is not alleviated by pleiotropy, but $n_{in}$ would need to exceed $n_{out}$ by an even wider margin if ion channels were not pleiotropic. Moreover, channels must be coadjusted with different ratios to regulate different properties, which constrains how feedback loops are organized (see below). These important issues get overlooked if regulation of each property is considered in isolation.

With respect to the number of channels required to coregulate a certain number of properties, a direct analogy can be made with a system of linear equations. Each unknown constitutes a degree of freedom and each equation constitutes a constraint that reduces the degrees of freedom by one. The system is said to be overdetermined if equations outnumber unknowns, and underdetermined if unknowns outnumber equations. An underdetermined system can have infinite solutions. Degeneracy is synonymous with underdetermination, which highlights that degeneracy depends not only on ion channel diversity, but also on those channels not having to coregulate 'too many' properties. Extra degrees of freedom broaden the range of available solutions, meaning a good solution is liable to exist over a broader range of conditions. On the other hand, if constraints outnumber the degrees of freedom, the system becomes overdetermined and solutions disappear (e.g., *Figure 5D*), which can cause regulation to fail (see below). One might reasonably speculate that ion channel diversity has been selected for because it enables the addition of new functionality (e.g., excitability) without the cell becoming overdetermined, lest pre-existing functions (e.g., osmoregulation) become compromised.

There are notable similarities and differences between a neuron adjusting its ion channel densities to regulate properties and a scientist trying to infer conductance densities based on the measured values of those properties (i.e., fitting a model to experimental data). Fitting a model with *n* parameters to many outputs (e.g., firing rate *and* input resistance *and* rheobase *and* spike height) is more difficult but yields better parameter estimates than fitting the same model to just one output (*Foster et al., 1993*), just as regulating more properties makes the solution set smaller. But how good are those parameter estimates? Assuming there is no measurement noise, can one confidently infer the true conductance densities in a particular neuron by measuring and fitting enough properties? Degeneracy makes solving this inverse problem difficult, if not impossible (*Sarkar and Sobie, 2010*; *Aster et al., 2013*). The neuron solves the forward problem, producing a firing rate, input resistance, etc. based on its channel density combination. That combination is determined by negative feedback but the precise densities are unimportant so long as the combination produces the target values for

all regulated properties (which happens for every density combination in the multioutput solution set). As conditions change, the multioutput solution will evolve and negative feedback will adjust densities accordingly. A neuron needs to regulate its properties but does not do so by regulating its conductance densities to particular values; in that respect, a neuron does not solve an inverse problem. (Convergence of conductance densities to the same 'attractive' set regardless of initial conditions, or after a perturbation, may suggest otherwise [*Franci et al., 2020*], but may reflect other, unaccounted for constraints like regulation of another property.) Neurons have evolved under selective pressure to be degenerate (see above), not parsimonious, contrary to how most models are constructed.

Previous studies using grid searches to explore degeneracy have tended to apply selection criteria simultaneously, finding the 'multioutput' solution set in one fell swoop. In contrast, we considered one criterion (property) at a time in order to find each single-output solution set, which we then combined to find the multioutput solution. The former approach is akin to aggregating several error functions to create a single-objective problem, whereas the latter resembles multiobjective optimization. Nevertheless, past studies have observed that certain parameter changes (in a particular direction through parameter space) dramatically affect some properties but not others (*Drion et al., 2015*; *Goldman et al., 2001*), consistent with manifolds representing the single-output solution sets for sensitive and insensitive properties lying orthogonal to one another in parameter space. The effects of solution dimensionality on ion channel correlations and regulation failure (see below) highlight the value of a multiobjective perspective, which

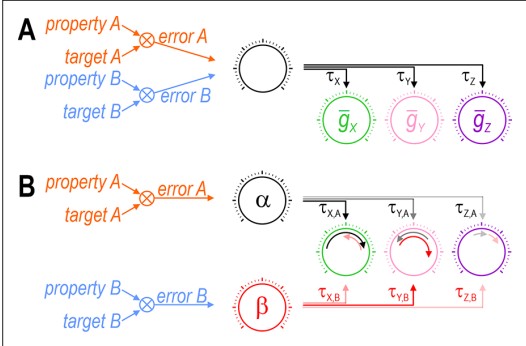

**Figure 11.** Each error signal must be able to coadjust channels with different ratios. Cartoons depict regulation of two properties (*A* and *B*) via control of three ion channels (*X*, *Y*, and *Z*). Channel levels are homeostatically adjusted to minimize the difference (error) between each property and its target value. The error signal divided by each channel's regulation time constant ($\tau$) dictates the magnitude of the change in that channel (represented by length of curved arrows in panel B). (**A**) The set of regulation time constants define a 'master regulator' (black dial) that coadjusts ion channels according to a certain ratio. Convergence of error signals at or before the master regulator limits how ion channels are coadjusted. (**B**) An additional master regulator (red dial) is needed to enable *error B* to coadjust channels with a different ratio than *error A*. Conditions like in panel B were required for all coregulation simulations (see *Figure 6C*). Different coadjustment ratios are required to regulate each property because of the different way each ion channel affects each property (see *Figure 1*). Note that master regulators may be more distributed than depicted here, with each regulation time constant $\tau_{i,j}$ likely reflecting the net effect of regulating multiple (transcriptional, translational, etc.) processes. The positive molecular regulatory network proposed by *Franci et al., 2020* involves additional feedback mechanisms not easily depicted in this sort of cartoon.

implicitly recognizes that there are multiple properties, each with its own feedback loop, even if those feedback loops intersect. Our results show that coregulating many properties using a common set of ion channels is feasible only if enough *different* channels are involved. Indeed, for coadjustments to work, channels must differ from each other in how they affect each property (see *Figure 1*), which emphasizes the distinction between degeneracy and redundancy.

Our results also highlight the need for >1 'master regulator' because regulation of each property requires that ion channels are coadjusted with different ratios (*Figure 11*). *O'Leary et al., 2014* argued in favor of a single master regulator, but that was predicated on using a single factor, namely global calcium, to encode the error signal; because a single factor cannot reach two different targets (i.e., simultaneously minimize two errors), it cannot support two master regulators. But local (*Bootman et al., 2001*) or kinetically distinct (*Liu et al., 1998*) variations in calcium could encode >1 error signal if the calcium sensors coupling intracellular calcium to gene expression or channel modulation are spatially segregated or operate with different filter properties. Beyond calcium, evidence points to firing rate homeostasis being activity independent (*MacLean et al., 2003*), using membrane potential as feedback (*Santin and Schulz, 2019*), or involving dual feedback loops (*Kulik et al., 2019*). Furthermore, the AMP:ATP ratio is used to track energy level (*Hardie, 2014*; *Garcia and Shaw, 2017*) and

free amino acids can also be sensed (*Efeyan et al., 2012*). This is just the tip of the iceberg. Rather than error signals being encoded by a single factor like calcium, we favor the notion of a multi-input/multioutput system (*Papin and Palsson, 2004*) in which factors combine to encode multiple error signals. Take mTOR (mechanistic target of rapamycin) for example; it is modulated by diverse factors and, in turn, modulates many processes including transcription, translation and protein degradation (*Switon et al., 2017*). The regulation model proposed by *Franci et al., 2020* provides additional insights. These issues require more investigation but, for now, our results argue that coregulation of multiple properties is incompatible with the amalgamation of error signals or certain control steps. The analogy with single- and multiobjective optimization (see above) seems apropos. Critically, $n_{in}$ exceeding $n_{out}$ will not guarantee solutions if degrees of freedom are reduced by bottlenecks elsewhere in the feedback loops.

Ion channel correlations have been studied in simulations (*O'Leary et al., 2014*; *Mukunda and Narayanan, 2017*; *Taylor et al., 2009*; *Jain and Narayanan, 2020*; *O'Leary et al., 2013*; *Franci et al., 2020*; *Soofi et al., 2012*; *Hudson and Prinz, 2010*; *Ball et al., 2010*) and experiments (*Zhao and Golowasch, 2012*; *Temporal et al., 2014*; *Schulz et al., 2006*; *Tobin et al., 2009*; *Schulz et al., 2007*; *Khorkova and Golowasch, 2007*). These correlations have been ascribed to the coregulation of ion channels. The relative rates with which different conductance densities are controlled dictate the direction in which trajectories move through parameter space, which in turn dictates how trajectories approach and distribute across the solution manifold (*O'Leary et al., 2013*). Our results are consistent with that explanation (*Figure 7*), notwithstanding noise effects (see below), but they also highlight the importance of manifold dimensionality: Correlations are stronger but are insensitive to relative regulation rates if trajectories reach a 1D manifold (curve) rather than a higher-dimensional manifold (surface, volume, etc.) (*Figure 6*). Under noisy conditions, correlations reflect the available solution space and how solutions drift across it, which depends on regulation rates (*Figure 8*). The cooperative molecular interactions proposed by *Franci et al., 2020* can produce correlations in a high-dimensional solution space, but may be prevented from doing so by a low-dimensional solution space. Notwithstanding molecular interactions, the existence of pairwise correlations suggests that the dimensionality of solution space (=$n_{in} - n_{out}$) is relatively low. This may be surprising since $n_{in}$ is high, but makes sense if $n_{out}$ is also high; in other words, there are *many* channels but they are responsible for regulating *many* properties. Though $n_{out}$ is constrained by $n_{in}$ (i.e., the number of regulated properties cannot exceed the number of adjustable ion channels), neurons might take advantage of regulating as many properties as their ion channel diversity safely allows, leading to a relatively low-dimensional solution space (as $n_{out}$ approaches $n_{in}$).

Homeostatic regulation can fail for different reasons. If there are no solutions (i.e., the solution set is empty), negative feedback cannot regulate a property to its target values. A multioutput solution set can be empty because single-output solution sets do not intersect (*Figure 10C*), meaning regulation of different properties to their respective target values is incompatible. Regulation can also fail because negative feedback fails to converge on available solutions (*Figure 9C*). Notably, a multioutput solution set may become less accessible (lower dimensional) as single-output solution sets start to separate (*Figure 10B*), and may foreshadow the eventual disjunction of those solution sets (*Figure 10C*). Regulation can also fail because feedback signaling is compromised. These failure modes are not mutually exclusive; for example, a system might reach less accessible solutions if regulation rates are normally flexible but might fail to reach those solutions if regulation rate flexibility is reduced. Cooperative molecular interactions (*Franci et al., 2020*) are notable in this regard, insofar as they would tend to constrain regulation and thus influence failure modes. Failed homeostatic regulation may have similar consequence regardless of how the failure occurs, consistent with the emergence of common disease phenotypes despite vastly different underlying pathologies (*Ramocki and Zoghbi, 2008*), but subtle differences in exactly how the failure transpires may provide important clues to help pinpoint the underlying mechanism(s).

In conclusion, neurons can coregulate multiple properties by coadjusting ion channels. Despite reusing channels to regulate more than one property, many channels are required to coregulate many properties and feedback loops must allow for coadjustments with different ratios. This may account for why evolution has yielded such diverse channels and why their transcriptional regulation has become so complicated (*Gabel et al., 2015*). Given how difficult it is to study regulation of any one property, studying the coregulation of multiple properties seems truly daunting, especially if feedback

loops intermingle and error signals involve combinatorial codes. But if that is indeed the case, then accounting for the coregulation of multiple properties may be crucial for appreciating design features that might otherwise elude us.

## Methods

### Slice electrophysiology

All procedures were approved by the Hospital for Sick Children Animal Care Committee. Using the same procedures and equipment previously described (*Khubieh et al., 2016*), coronal slices of hippocampus were prepared from an adult mouse and a CA1 pyramidal neuron was recorded using whole-cell patch clamp. A junction potential correction of −9 mV was applied to the recorded membrane potential. A noisy stimulus was generated through an Ornstein–Uhlenbeck process (see *Equation 7*) with $\tau_{stim}$ = 5 ms, $\mu_{stim}$ = 60 pA, and $\sigma_{stim}$ = 10 pA; the same stimulus was replayed on each trial. Native $I_M$ was blocked by bath application of 10 µM XE991 (Tocris). The block was continued throughout dynamic clamp experiments. Virtual $I_M$ and $I_{AHP}$ were modeled as per *Equation 6* (with $\tau_z$ = 100 ms, $\beta_z$ = −35 mV, and $\gamma_z$ = 4 mV for $I_M$, and $\tau_z$ = 600 ms, $\beta_z$ = 0 mV, and $\gamma_z$ = 1 mV for $I_{AHP}$) and were applied using the dynamic clamp capabilities of Signal v6 (Cambridge Electronic Design). The density of each virtual conductance was adjusted to produce the desired firing rates ($\bar{g}_M$ = 10 nS, $\bar{g}_{AHP}$ = 120–150 nS).

### Neuron model

Our base model includes a fast-activating sodium conductance ($g_{fast}$) and a slower-activating potassium conductance ($g_{slow}$) which together are sufficient to produce spikes. Other conductances were added to modulate excitability.

$$C\frac{dV}{dt} = I_{stim} - \bar{g}_{fast}m_\infty(V)(V - E_{Na}) - \bar{g}_{slow}w(V - E_K) - \bar{g}_{Na}n(V - E_{Na}) -$$
$$\bar{g}_K n(V - E_K) - \bar{g}_{aHP}p(V - E_K) - \bar{g}_M q(V - E_K) - \bar{g}_{leak}(V - E_l), \tag{1}$$

where $V$ is voltage and $m$ changes instantaneously with $V$ whereas other gating variables change more slowly. The spike-generating conductances $g_{fast}$ and $g_{slow}$ were modeled using a Morris–Lecar formalism (*Rho and Prescott, 2012*), with $C$ = 2 µF/cm$^2$, $E_{Na}$ = 50 mV, $E_K$= −100 mV, $E_{leak}$ = −70 mV, $\phi_w$= 0.15, $\bar{g}_{fast}$ = 20 mS/cm$^2$, $\bar{g}_{slow}$ = 20 mS/cm$^2$, $\bar{g}_{leak}$ = 2 mS/cm$^2$, $\beta_m$ = −1.2 mV, $\gamma_m$ = 18 mV, $\beta_w$ = −10 mV, and $\gamma_w$ = 10 mV. A generic sodium conductance ($g_{Na}$) and potassium conductance ($g_K$) were modeled using Hodgkin–Huxley formalism as described by *Ratté et al., 2014b*

$$I_{Na,K} = \bar{g}_{Na,K}n(V - E_{Na,K}), \tag{2}$$

$$\frac{dn}{dt} = \alpha(1 - n) - \beta n, \tag{3}$$

$$\alpha = \frac{k_\alpha(V - V_\alpha)}{e^{(V - \frac{V_\alpha}{s_\alpha})} - 1}, \tag{4}$$

$$\beta = k_\beta e^{\left(V - \frac{V_\alpha}{s_\alpha}\right)}, \tag{5}$$

where $V_{\alpha,\beta}$ = −24 mV, $s_{\alpha,\beta}$ = −17 mV, and $k_{\alpha,\beta}$ = 1 ms$^{-1}$. Note that $g_{Na}$ and $g_K$ differ only in their reversal potentials. Channels approximating a calcium-activated potassium conductance ($g_{AHP}$) and an M-type potassium conductance ($g_M$) were modeled as described by *Prescott and Sejnowski, 2008*

$$\frac{dz}{dt} = \left\{ \frac{1}{1 + e^{\left(\frac{\beta_z - V}{\gamma_z}\right)}} - z \right\} / \tau_z, \tag{6}$$

where $\tau_z$ = 100 ms, $\gamma_z$ = 4 mV, and $\beta_z$ = 0 mV or −35 mV for $g_{AHP}$ and $g_M$, respectively. Maximal conductance densities for $g_{leak}$, $g_{Na}$, $g_K$, and $g_M$ were systematically varied or adjusted by a homeostatic feedback mechanism (see below). Any other parameters that differ from the sources cited above are reported in the relevant figure legends. Injected current $I_{stim}$ was applied as either a constant step or as noisy fluctuations modeled with an Ornstein–Uhlenbeck process

$$\frac{dI_{stim}}{dt} = \frac{I_{stim} - \mu_{stim}}{\tau_{stim}} + \frac{S\sigma_{stim}}{\sqrt{dt}}N(t), \tag{7}$$

where $\tau_{stim}$ is a time constant that controls the rate at which $I_{stim}$ drifts back toward the mean $\mu_{stim}$, and $N(t)$ is a random number drawn from a normal distribution with 0 mean and unit variance that is scaled by $S\sigma_{stim}$, where a scaling factor $S = \sqrt{2/\tau_{stim}}$ makes the standard deviation $\sigma_{stim}$ independent of $\tau_{stim}$. All simulations were conducted in MATLAB using the forward Euler integration method and a time step of 0.05–0.1 ms.

## Energy calculations

Energy consumption rate was calculated as in *Hasenstaub et al., 2010* Briefly, models were stimulated with a fast fluctuating stimulus ($\mu_{stim}$ = 40 µA/cm², $\sigma_{stim}$ = 10 µA/cm²). Sodium and potassium current through all channels was integrated for 1 s to determine the charge for each ion species, which was then converted to ion flux based on the elementary charge $1.602 \times 10^{-19}$ C. Based on the 3:2 stoichiometry of the Na$^+$/K$^+$ pump, we divided the number of sodium and potassium ions by 3 and 2, respectively, and used the maximum of those two values as the energy consumption rate (in ATP/cm² s). Energy efficiency was calculated as the ratio between capacitive minimum and total Na$^+$ flux during an action potential (*Sengupta et al., 2010*). Briefly, voltage was reset to −40 mV to evoke a single spike in all models. Models were treated as pure capacitors to calculate the minimum capacitive current as $C\Delta V$, where $C$ is the capacitance and $\Delta V$ is the difference between the resting membrane potential and the spike peak.

## Grid search

Models were tested with conductance density combinations chosen from a 100 × 100 or 30 × 30 × 30 grid for 2D and 3D plots, respectively. To depict each single-output solution set, all models with outputs within a range (tolerance) of the target value (±3 spk/s for firing rate, ±0.25% for energy efficiency, and ±0.003 kΩ cm² for input resistance) were selected and a curve or surface was fit to those successful models. The same tolerances were implemented in the homeostatic learning rule (see below) to minimize ringing. Tolerances are illustrated in *Figure 5—figure supplement 1* but were not shown in other figures for sake of clarity.

## Feedback control

We used a homeostatic learning rule similar to *O'Leary et al., 2014*; *O'Leary et al., 2013* with two notable differences: (1) we did not use intracellular calcium or other biological signals (e.g., AMP:ATP ratio) as intermediaries for our error signals and (2) the error for each output was determined as the difference between the current and target values at the end of each iteration, and conductance densities were adjusted before the start of the next iteration, rather than updating and feeding back error signals in 'real time'. The separation between fast and slow (regulatory) timescales justifies the former approach. For each conductance $\bar{g}_i$, error was divided by the regulation time constant ($\tau_i$) and added to the conductance (see *Figure 4A*). For coregulating >1 property, we used the sum of scaled errors to update conductance densities (see *Figure 6D*). A single run consists of a maximum of 200 iterations, during which a model must reach and maintain its regulated property within the tolerance for five consecutive iterations. All models that reached the target output(s) did so in well under 100 iterations; regulation was deemed to have failed for models not reaching their target output(s) within 200 iterations. Conductance densties during the last five iterations were averaged and reported as the final value. See *Supplementary file 1* for the initial conductance densities and regulation time constants used for each figure.

## Conductance noise

For simulations in *Figure 8*, noisy variations in each conductance density, $\bar{g}_{Na}$, $\bar{g}_K$, and $\bar{g}_M$, was applied by adding a random number drawn from a Gaussian distribution with a mean of 0 and a standard deviation of 0.05 mS/cm². Noise was independent for each conductance. After applying noise, the simulation was run, errors were calculated, and conductance densities were updated according to the feedback control described above; this qualifies as one noise-update iteration. Up to 3000 noise-update iterations were run. If the addition of noise caused a conductance density to become negative, the density was reset to 0 mS/cm² before applying feedback control. Where indicated, conductance density was likewise reset to 4 mS/cm² if noise caused it to increase above that value.

## Code availability
All computer code is available at http://modeldb.yale.edu/267309 and at http://prescottlab.ca/code-for-models.

## Acknowledgements
This research was funded by a Discovery Grant from the Natural Sciences and Engineering Research Council (NSERC) and by a Foundation Grant from the Canadian Institutes of Health Research (CIHR). H.S. was supported by Ontario Graduate Scholarship, the University of Toronto Centre for the Study of Pain Scholarship, and James F Crothers Family Scholarship. J.Y. was supported by a SickKids Restracomp Studentship and Ontario Graduate Scholarship. We thank Eve Marder, Etay Hay, Shreejoy Tripathy, and Simon Hardy for constructive feedback on the manuscript.

## Additional information

### Funding

| Funder | Grant reference number | Author |
|---|---|---|
| Canadian Institutes of Health Research | Foundation Grant 167276 | Steven Alec Prescott |
| Natural Sciences and Engineering Research Council of Canada | Discovery Grant RGPIN 436168 | Steven Alec Prescott |

The funders had no role in study design, data collection, and interpretation, or the decision to submit the work for publication.

### Author contributions
Jane Yang, Conceptualization, Data curation, Formal analysis, Investigation, Visualization, Writing – original draft, Writing – review and editing; Husain Shakil, Conceptualization, Data curation, Formal analysis, Investigation, Writing – review and editing; Stéphanie Ratté, Data curation, Formal analysis, Investigation, Writing – review and editing; Steven A Prescott, Conceptualization, Funding acquisition, Methodology, Project administration, Supervision, Writing – original draft, Writing – review and editing

### Author ORCIDs
Jane Yang (ID) http://orcid.org/0000-0003-0114-5503
Husain Shakil (ID) http://orcid.org/0000-0003-3995-6811
Stéphanie Ratté (ID) http://orcid.org/0000-0002-7005-6726
Steven A Prescott (ID) http://orcid.org/0000-0002-3827-4512

### Ethics
All experimental procedures were approved by The Hospital for Sick Children Animal Care Committee (protocol #53451) and were conducted in accordance with guidelines from the Canadian Council on Animal Care.

### Decision letter and Author response
Decision letter https://doi.org/10.7554/eLife.72875.sa1
Author response https://doi.org/10.7554/eLife.72875.sa2

## Additional files

### Supplementary files
• Supplementary file 1. Initial conductance densities and regulation time constants used for simulations. When a conductance is not regulated, its density was fixed at the average.

• Transparent reporting form

## Data availability

All computer code is available at http://modeldb.yale.edu/267309 and at http://prescottlab.ca/code-for-models. Key parameter values are provided in Supplementary file 1. Other parameter values are identified in the Methods. Source data are provided for Figure 2.

The following dataset was generated:

| Author(s) | Year | Dataset title | Dataset URL | Database and Identifier |
|---|---|---|---|---|
| Yang J | 2022 | A modified Morris-Lecar model with gM and gAHP | http://modeldb.yale.edu/267309 | ModelDB, 267309 |

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
