## [Editor Report]

Neurons develop and maintain rich electrophysiological properties that enable nervous systems to function. The question of how different neural properties are regulated by internal ion channel expression mechanisms remains unresolved. In this paper, Yang and colleagues address the question from an abstract perspective by asking how multiple constraints on the physiological properties of neurons, such as firing rate curves and energy efficiency, narrow down the available regulatory possibilities. Their results from a mixture of modelling and dynamic clamp experiments point to the existence of multiple parallel internal feedback loops for controlling ion channel expression in neurons, and derive conditions under which co-regulation mechanisms will fail.

---

## [Decision Letter]

**Decision letter after peer review:**

Thank you for submitting your article "Minimal requirements for a neuron to co-regulate many properties and the implications for ion channel correlations and robustness" for consideration by *eLife*. Your article has been reviewed by 2 peer reviewers, and the evaluation has been overseen by a Reviewing Editor and John Huguenard as the Senior Editor. The following individual involved in review of your submission has agreed to reveal their identity: Alessio Franci (Reviewer #1).

Essential revisions:

1) Return to the analysis and modeling to address whether the results may substantially alter depending on the regulation model assumed. In particular, take account of the updated mathematical model described in Franci et al., (2020) IEEE Control Systems Letters 4 (4), 946-951.

2) Address concerns about the experimental data. In particular, stronger statistical support is needed regarding reported changes in spike height and resting membrane potential and the reversal of these changes with dynamic clamp. In addition, please provide a better example neuron for Figure 2, which seems to show a very depolarized resting potential of approximately -30 mV after blocking I-m. This level of depolarization is inconsistent with any spiking at all.

3) Address the specific comments detailed in the reviewer assessments below.

*Reviewer #1 (Recommendations for the authors):*

I really appreciated the effort of this paper but the modeling assumptions underlying it hamper its biological relevance. My suggestion would be to redo the analysis when the regulation loops are not independent (in particular, when the target for one loop depends on the value of all the other variables) and when the revised model in Franci et al., 2020 is used for each loop.

*Reviewer #2 (Recommendations for the authors):*

I strongly suggest changing the way the solution surfaces are displayed in Figures 6 – 9. The hatching patterns make it difficult to see other elements of the figure, such as the dots and regulation trajectories. The hatching also does not add anything to the perception how the surface is situated in parameter space. For example, in Figure 5 supplement 1 the surfaces are easy to see without hatching.

---

## [Author Response]

Essential revisions:1) Return to the analysis and modeling to address whether the results may substantially alter depending on the regulation model assumed. In particular, take account of the updated mathematical model described in Franci et al., (2020) IEEE Control Systems Letters 4 (4), 946-951.

According to our understanding of Franci et al., (2020), that study was prompted in part by the observation that ion channel correlations emerging through the regulation mechanism used by O’Leary et al., (PNAS 2013, Neuron 2014) are disrupted by noise. Noise is ubiquitous and we totally agree that noise effects must be considered (see below). Other considerations like whether conductance densities converge on the same values regardless of initial conditions (i.e. whether the solution set is attractive) may influence the choice of regulation models in certain contexts (e.g. emergence of cell types during development; O’Leary et al., 2014) but we do not think this is critical for the regulation we studied; for instance, we are not aware of strong experimental evidence that conductance densities necessarily return to their original values after a perturbation. Thus, the most important question, it seems to us, is whether our conclusions are invalidated by noise.

To investigate this, we added noise to our model and explored the effects on ion channel correlations under different conditions (i.e. for solution spaces with different dimensionality). By adding noise to each conductance density and allowing feedback regulation to restore output(s) to target value(s), we show that solutions spread out across the solution manifold, consistent with Franci et al., (2020). This is now shown in Figure 8, which is an entirely new figure on the effects of noise. When the solution space is low-dimensional (when a submanifold is formed by intersecting manifolds), none of the conductance densities can rise very high without one of the other conductance densities hitting zero (Figure 8C) and the spread is thus bounded. As solutions spread across the submanifold, correlations strengthen but are not qualitatively altered. For higher-dimensional solution spaces, solutions drift across the manifold and sometimes proceed outside the illustrated region (Figure 8A). But saturation of rate-limiting production steps almost certainly occurs, and would limit how high conductance densities can rise (lines 657-659); the effect of including an upper bound is shown in Figure 8—figure supplement 1. These bounds prevent infinite spread. Noise-induced spreading of solutions on a higher-dimensional solution manifold destroys the correlations formed according to the O’Leary et al., (2013) mechanism, but other correlations emerge, as explained in Figure 8 and its supplement. These results are exciting insofar as they affirm our main conclusion about the importance of solution space dimensionality: effects of noise on ion channel correlations depend on the dimensionality of the solution space.

Our new results also demonstrate that our simple regulation mechanism can successfully co-regulate multiple properties under noisy conditions, and can do so without feedback loops being coupled. This is all that we asked of the regulation mechanism insofar as we did not set out to explain the emergence of ion channel correlations or other such phenomena; instead, we focused on whether such outcomes differed depending on the number of regulated properties and the number of adjustable ion channels. That said, outcomes might differ depending on the regulation mechanism (see below). But we do not think it is fair to impose additional requirements on the regulation mechanism that are not essential for our focus (e.g. that the solution set is attractive). Whether a neuron regulates its properties or its conductance densities is addressed at some length in our Discussion (lines 296-315).

We strove to implement the cooperative molecular interactions proposed by Franci et al., but ran into difficulties because we do not fully understand the mathematical explanations in the 2020 paper. Apart from these technical difficulties, we also have reservations about certain suggestions. For example, coupling the feedback loops in the proposed manner requires use of a common feedback signal, which is something we believe does not happen (see Lines 342-344) and is unnecessary from our perspective (see above). Our own unpublished experimental data make us skeptical about the premise for cooperative interactions; specifically, we have found that different sodium channel types are negatively correlated and that this is not reflected in transcript levels because it depends on the regulation of translation, not transcription. There is clearly much more work to do in this area, but insofar as our model provides valuable insights using a feedback mechanism that does not have obvious shortcoming (including under noisy conditions), exploring alternative feedback mechanisms arguably exceeds the scope of the current study.

Though we did not get cooperative molecular interactions working in our model, we have added discussion on how anticipated results would differ from the results obtained with our simple regulation mechanism. In a high-dimensional solution space, we expect that cooperative interactions would preclude correlations arising from the mechanism shown in Figure 8A since the Franci mechanism prevents solution from drifting (explained on lines 226-227) and would, instead, produce other correlations. On the other hand, in a low-dimensional solution space, the constraints of the solution space (Figure 8C) may prevent correlations unless the attractive subspace emerging from molecular interactions align with the solution space (lines 227-229). This is addressed also in the Discussion (lines 362-364). We have also touched upon how other issues addressed by Franci et al., (2020) apply to our results, including attractiveness of the solution set (lines 312-314) and how cooperative molecular interactions might influence success of regulation (lines 242-243; 381-383).

2) Address concerns about the experimental data. In particular, stronger statistical support is needed regarding reported changes in spike height and resting membrane potential and the reversal of these changes with dynamic clamp. In addition, please provide a better example neuron for Figure 2, which seems to show a very depolarized resting potential of approximately -30 mV after blocking I-m. This level of depolarization is inconsistent with any spiking at all.

We have analyzed the experimental data as requested and have incorporated this analysis into Figure 2. Statistical analysis has been reported in detail in the figure legend (lines 511-524). With respect to membrane potential, a junction potential correction had not been applied to data reported in the original submission; applying the -9 mV correction (lines 400-401) renders the values more accurate. Abrupt blockade of I_m_ is expected to cause strong depolarization which will in turn cause some sodium channel inactivation and attenuated spike height. That inactivation is relieved by the hyperpolarization produced by either of the two virtual conductances, despite continued presence of the drug, 4-AP. In short, the chosen traces exemplify what we are trying to illustrate.

We want to clarify here that this experiment was conducted in a single neuron. Certain comments suggest that this was not clear, and we have revised the text to avoid potential confusion. We did not repeat the experiment in multiple neurons because our goal was not to quantify the impact of the compensatory mechanism (which depends on the strength of the virtual conductance we insert); rather, we sought simply to show that different cell properties are differently affected by each compensatory mechanism (as proposed in Figure 1), and that such differences are not an artifact of our model neuron, which is why we conducted this initial test experimentally rather than with simulations. Insofar as we would apply the same virtual conductance if we were to repeat the experiment in additional neurons, we would not expect notable differences. Trials were repeated to assess spike timing across trials rather than with statistical analysis in mind, but effects are evidently very reproducible across trials.

3) Address the specific comments detailed in the reviewer assessments below.

We have responded to all specific comments in the reviewer assessments below.

Reviewer #1 (Recommendations for the authors):I really appreciated the effort of this paper but the modeling assumptions underlying it hamper its biological relevance. My suggestion would be to redo the analysis when the regulation loops are not independent (in particular, when the target for one loop depends on the value of all the other variables) and when the revised model in Franci et al., 2020 is used for each loop.

Thank you for the detailed suggestions. We hope the responses we provided above have addressed many of the concerns. We will address additional issues here, including our attempts to introduce cooperative molecular interactions and coupled feedback loops into our model.

Without solid experimental data for many aspects of our model, we tried to minimize assumptions by keeping our model simple (see lines 79-90). This extends to using minimalist feedback loops (i.e. not implicating specific biophysical signals or specific transcriptional/ translational/ post-translational processes). This limits the scope of our conclusions but, on the other hand, this makes us more confident in the conclusions we have drawn and provides a solid foundation for future work. We think it is reasonable to start by assuming that feedback loops are independent and we believe that experimental data support their independence (see above), and so it is valuable to know whether such an arrangement can produce ion channel correlations (with or without noise). This is now addressed in Figure 8.

As explained under point 1 of the essential revisions, we cannot follow all the mathematical arguments presented by Franci et al., (2020). This complicated our efforts to implement cooperative molecular interactions in our model. Combined with our reservations about the impetus for implementing certain changes (as expressed above), we stopped short of testing a fundamentally different regulation mechanism as part of the current study. As evident from a number of elegant studies motivated by and published since the 2014 O’Leary paper in Neuron, we believe that the simple regulation mechanism they proposed can lead to many valuable insights. Our study continues in that vein. We are intrigued by the regulation mechanism proposed by Franci et al., and we think it requires further investigation in connection with some of the concepts raised in our study, but we also think certain concerns about unrobustness must be tempered based on the focus of our study. We welcome deeper discussions with the reviewers, unfettered the peer review process, to push this work forward collaboratively. We also think there is an opportunity for Franci to write a piece to accompany our article, especially if he can implement his regulation mechanism in our model and show simulation results that challenge or support our conclusions.

Reviewer #2 (Recommendations for the authors):I strongly suggest changing the way the solution surfaces are displayed in Figures 6 – 9. The hatching patterns make it difficult to see other elements of the figure, such as the dots and regulation trajectories. The hatching also does not add anything to the perception how the surface is situated in parameter space. For example, in Figure 5 supplement 1 the surfaces are easy to see without hatching.

We have lightened the hatching on all solution surfaces. When we removed the hatching, we realized that the hatching helped give some perspective. That perspective can be achieved with light hatching that does not obscure the solutions or their trajectories. We think our changes serve to optimize visualization.